# Characterizing Spray-Dried Powders through NIR Spectroscopy: Effect of Two Preparation Strategies for Calibration Samples and Comparison of Two Types of NIR Spectrometers

**DOI:** 10.3390/foods12030467

**Published:** 2023-01-19

**Authors:** Zhiyang (Stan) Tu, Joseph Irudayaraj, Youngsoo Lee

**Affiliations:** 1Department of Food Science and Human Nutrition, University of Illinois at Urbana-Champaign, Urbana, IL 61801, USA; 2Department of Bioengineering, University of Illinois at Urbana-Champaign, Urbana, IL 61801, USA

**Keywords:** spray drying, NIR spectroscopy, PLS regression, food composition analysis, process optimization

## Abstract

Emerging portable near infrared (NIR) spectroscopic approaches coupled with data analysis and chemometric techniques provide opportunities for the rapid characterization of spray-dried products and process optimization. This study aimed to enhance the understanding of applying NIR spectroscopy in spray-dried samples by comparing two sample preparation strategies and two spectrometers. Two sets of whey protein–maltodextrin matrixes, one with a protein content gradient and one with a consistent protein content, were spray-dried, and the effect of the two preparation strategies on NIR calibration model development was studied. Secondly, a portable NIR spectrometer (PEAK) was compared with a benchtop NIR spectrometer (CARY) for the moisture analysis of prepared samples. When validating models with the samples with focused protein contents, the best PLS protein models established from the two sample sets had similar performances. When comparing two spectrometers, although CARY outperformed PEAK, PEAK still demonstrated reliable performance for moisture analysis, indicating that it is capable as an inline sensor.

## 1. Introduction

Drying is one of the crucial steps in food and ingredient processing given its utility in extending shelf life and improving the functionality of ingredients. Spray-drying has been found to be a highly versatile drying method for the transformation of a liquid feed to a powder. Spray-drying produces powders with a low moisture content and a high bulk density, which results in storage stability and a reduced cost of packaging and transportation [1]. Due to good commercial and economic feasibility, the industrial applications of spray-drying have expanded to a wide range of powder products including dairy and beverage products [2,3]. The emerging trend in using spray-dried microencapsulation to protect heat-sensitive and bioactive compounds has led to the further use of spray drying for oils [4] and flavors [5]. Whey protein isolate (WPI) and maltodextrin are commonly used as wall materials for the preparation of microcapsules due to their low costs and wide availability [4]. A rapid method to characterize WPI and maltodextrin would enhance our understanding of the microencapsulation process and improve the qualities of microcapsules.

Process analytical technology (PAT) is an initiative led by the Food and Drug Administration (FDA) that promotes the development of in-process sensing technologies to improve manufacturing control, enhance process understanding, improve product quality, and reduce waste [6]. The food industry has adapted this approach by using multivariate measurements and chemometric analysis for the process monitoring and quality control of dried apple chips [7,8], honey [9], starches [10], infant formula [11,12], dairy products [13], citrus microcapsules [14], protein powder [15], and edible oils [16,17]. Among the instruments applied in PAT solutions, near infrared spectroscopy (NIR) has been the most popular analyzer because NIR requires no sample preparation, providing possibilities for on-line or in-line rapid measurements [18]. The recent emergence of low-cost portable miniature NIR spectrometers has further expanded the popularity of NIR. Portable spectrometers provide excellent flexibility in the rapid and non-destructive analysis of a drying process, which is not possible with bench top spectrometers [19]. The quality of spray-dried products depends on the feed characteristics as well as the drying conditions such as the liquid feed rate, drying gas flow rate, inlet and outlet temperature, etc. [20,21]. A PAT solution for spray drying could greatly simplify the process control by providing rapid feedback of the product quality, shortening the troubleshooting process and reducing defective products and thus reducing waste and energy cost.

NIR spectroscopy is based on the Beer–Lambert Law. The chemical vibrations and their overtones and combinations share a linear relationship with the absorbance of light at certain wavelengths (feature bands). The wavelength region of NIR includes overlapped feature bands of common chemical vibrations in foods. Through chemometric tools such as partial least square regression (PLS), the chemical information in NIR spectra can be extracted to calibrate prediction models. A typical procedure for establishing NIR prediction models includes sample preparation, sample characterization, model calibration, and model validation. Calibration samples are the foundation of prediction models. Calibration algorithms establish models by correlating the chemical composition of calibration samples to corresponding NIR spectra. To validate the model performance, the models predict the composition of validation samples and compare the results to reference values acquired through conventional analysis. PLS is the most popular tool to establish NIR prediction models [22] and has been widely applied in analysis of agricultural and biological materials. The chemical compositions of agricultural samples vary slightly across samples, and a large number of samples (>80) is often required to establish a robust linear relationship between compositions and the spectra [23,24]. This sample preparation approach can be referred to as a local set. For formulated food samples, it is possible to design a calibration sample set with a wide distribution of certain compositions, which can be referred to as a global set. Models built on a global set have a greater working range than those built on a local set, meaning that the prediction results are more accurate when it comes to outliers. A global set could also decrease the sample set size required for a robust model, as the linear relationship of the analyte and the spectrum is strong in contrast to the spectral noise.

Although portable NIR spectrometers have become a valuable resource in composition analysis, and NIR spectroscopy has been applied for the characterization of spray-dried products, to the best of the authors’ knowledge, no investigation exists on the characterization of WPI–maltodextrin matrixes through NIR. When characterizing spray-dried samples through NIR spectroscopy, there are several challenges. The differences in the particle size and morphology of spray-dried powders lead to undesired interference in NIR analysis [21]. The variation of the homogeneity of sample characteristics between batches also adds complexity to the application of NIR in spray-drying. Moreover, the preparation of a large number of calibration samples is time consuming and labor intensive. The overall goal of this study was to enhance the understanding of applying NIR spectroscopy in spray-dried samples. Specifically, this study aims to compare NIR protein models built on a heterogeneous set of spray-dried WPI–maltodextrin matrixes with various protein contents (global set) and a homogeneous set with a focused protein content (local set). This study also aims to compare the performance of a portable NIR spectrometer for moisture analysis with a benchtop NIR spectrometer. In addition, several chemometric optimization techniques (preprocessing and data partition) were evaluated to enhance the robustness of the prediction models.

## 2. Materials and Methods

### 2.1. Materials

Hilmar™ 9000 Whey protein isolate was supported by Hilmar Ingredients (Hilmar, CA, USA). Maltodextrin (DE10) and N-OSA modified starch (CAPSUL^®^) were provided by Ingredion (Westchester, IL, USA). 

### 2.2. Sample Preparation

The feed solutions were prepared by dissolving 100 g of solid mixture into 400 g of deionized water to achieve a 20% (*w*/*w*) solid concentration. The solid mixture included maltodextrin DE 10 (MD10), modified starch, and whey protein isolate (WPI). The mass ratio of MD10 and modified starch remained at 4:1 across samples, and the WPI content varied depending on the experimental design. Two sets of samples were prepared in order to compare the global and local approaches (two protein content distributions of samples for NIR calibration modeling). As shown in Table 1, set A (global set) comprised 16 calibration samples with WPI contents varying from 10% to 25% (*w*/*w*) of the total solid with a 1% increment. Thereafter, five validation samples were spray-dried with approximate WPI contents of 12%, 15%, 18%, 21%, and 24% (*w*/*w*). On the other hand, Set B (local set) comprised 20 spray-dried samples, each with approximately 15% (*w*/*w*) WPI content. Overall, a total of 41 whey protein–maltodextrin matrixes were prepared and spray-dried in this study.

### 2.3. Spray Drying

A lab-scale spray-dryer (BÜCHI Mini Spray Dryer B-290, Flawil, Switzerland) was used for the preparation of samples. The inlet temperature was 160 °C and the outlet temperature was controlled at 85~90 °C by adjusting the feed rate from 4.5 to 7.5 mL/min. The aspirator flow rate was 35 m^3^/h, and the spray gas flow rate was 667 L/h. Right after spray drying, powders were transferred into glass (2 oz. clear glass straight-sided squat jar, Qorpak, PA, USA), sealed, and stored at room temperature.

### 2.4. Protein Analysis

A combustion nitrogen analyzer, rapid N exceed^®^ (Elementar, Ronkonkoma, New York, NY, USA), was used for the protein analysis. In total, 250 mg aspartic acid was used as the blank and triplicates of 250 mg sample were loaded into the equipment. The nitrogen to protein factor was 6.25.

### 2.5. Moisture analysis

A Mettler Toledo HR83-P Halogen Moisture Analyzer (Columbus, OH, USA) was used to carry out the moisture analysis. Approximately 1 g of sample was loaded onto an aluminum plate, and the drying temperature was set at 90 °C. The moisture content is expressed as % w.b. (wet basis).

### 2.6. Acquisition of NIR Spectra

Two types of NIR equipment were used in this study for characterization of spray-dried powders. A portable NIR spectrometer was employed to collect NIR spectra of samples in a jar. The spectrometer included a miniature spectrometer module, PEAK XNIR (Ibsen Photonics A/S, Farum, Denmark), a tungsten–halogen fiber light source (ASB-W-020, Spectral Products, Putnam, CT, USA), and an NIR reflectance probe (R600, StellarNet Inc., Tampa, FL, USA). The InGaAs detector in the spectrometer module was cooled to −20 °C, and the exposure time was set at 20.32 ms. The scan mode was set to Hadamard. The spectra comprised 214 bands between 1650 nm to 2400 nm. Triplicate scanning was performed, and the samples were shaken thoroughly between the replicates to ensure homogeneity.

A benchtop NIR spectrometer (Varian CARY 5G, Agilent Technologies, Inc., Santa Clara, CA, USA) was also used to collect NIR spectra in the diffuse reflectance mode. Samples were loaded onto a disk of the diffuse reflectance accessory (Cricket™, Harrick Scientific Products, Inc., New York, NY, USA), and the sample surface was evened by a flat spatula. Zero/baseline correction was performed. Duplicate scanning was carried out with 1600 bands between 900 nm and 2500 nm. The resolution of the spectrometer was 1 nm. The spectral data acquired by both spectrometers were then analyzed using MATLAB (Version R2021b, The Mathworks, Natick, MA, USA).

### 2.7. Spectral Analysis

#### 2.7.1. Preprocessing

Preprocessing is a necessary step in spectral analysis. Spectral variations and baseline shifts often exist in NIR spectra due to instrumental noise, changes in sample morphology and density, differences in the path length between spectrum scanning, and the use of fiber. To decrease the impact of the undesired interference, Savitzky–Golay filtering (SG), standard normal variate transformation (SNV), and first derivative were used. SG reduces instrumental noise by smoothening NIR spectra through polynomial fitting [25,26]. A second-degree SG filter with a 25-point window was applied to the spectra of the set A samples, and a third-degree filter with a 9-point window was used for the set B samples. SNV was used to reduce vertical baseline shift [27]. The derivative method, on the other hand, is effective in reducing additive and multiplicative baseline variations.

#### 2.7.2. Data Partition

To simulate the preparation of external validation samples, sample set A included validation samples that were spray-dried following the completion of calibration samples. However, when external validation samples are unavailable, dividing the existing set of samples into a calibration set and validation set could also be used to validate the robustness of the calibration models [28]. For the protein model calibration, sample set B was divided into calibration sets and validation sets through systematic sampling, the Kennard–Stone algorithm (KS), and the Joint XY (SPXY) algorithm. Systematic sampling first ranked the samples based on their y values (protein, moisture, etc.), and every fourth or third sample was selected as a validation sample. The unselected samples became the calibration set. The KS algorithm selects samples based on their distance in the instrument response data space and has been widely used in NIR spectroscopy for data partition [29,30,31,32,33]. The SPXY algorithm selects samples in a similar way as KS algorithm but focuses on the distance in both the instrumental response space and the dependent variable space to capture more information regarding the relationship of the samples [34]. The samples for moisture models were selected in a similar manner as before. 

#### 2.7.3. Development of the Calibration Models

In this study, leave-one-out cross validation was performed to determine the optimal number of latent variables to avoid overfitting. The model robustness was evaluated using the coefficient of determination for calibration (R^2^_c_) and for validation (R^2^_v_), root mean square error of calibration (RMSEC), and root mean square error of validation (RMSEV). The variable importance in the projection (VIP) [35] for each model was computed for the interpretation of NIR spectra.

#### 2.7.4. Comparison of Global and Local Protein Models

To establish NIR calibration models, a large number of samples is usually required for a robust model, and it is a common practice that the samples used in model calibration have a small range of y values. For formulated samples such as spray-dried powder, it is possible to design a sample set that includes samples with evenly distributed y values within a range to establish a global calibration model. It was hypothesized that a global sample set can calibrate a robust NIR model with a limited number of samples due to the additivity of the dependent variable and the instrumental response. To test this hypothesis, a series of protein calibration models were established based on sample sets A and B. In set A, 16 calibration samples were used to establish protein calibration models with three different preprocessing methods. Those models were validated by 5 validation samples in set A. In set B (excluding one outlier), 13 samples were selected through rank systematic sampling, KS, and SPXY to establish calibration models. The remaining 6 samples in set B were used to validate the performance of the models internally. Among the models established through different data partition methods, KS and SPXY data partition algorithms have shown performance advantages compared to systematic sampling in the validation phase. Thus, KS and SPXY were used to select 13 samples from 16 calibration samples in set A to establish calibration models, which were validated by the same samples that were used to validate set B models. Then, all 21 set A samples were used to establish a calibration model which was validated by all 20 set B samples.

#### 2.7.5. Comparison of a Benchtop NIR Spectrometer and a Portable NIR Spectrometer

A reflectance NIR spectrometer (PEAK) for spray-dried powders was developed to characterize the moisture content of spray-dried powders. To evaluate the effectiveness of scanning through a glass jar with this NIR spectrometer in the wavelength range from 1650 nm to 2400 nm, spectra were acquired from the spray-dried samples by PEAK and a benchtop NIR spectrometer (CARY). Combining set A and set B, a total of 82 spectra were collected from the two spectrometers. The spectra were preprocessed by SNV+SG and first derivative, respectively, before being divided by systematic sampling (Sys), KS, and SPXY, respectively, where 28 samples were selected as the calibration set and the remaining 13 as the validation set. Moisture models with various combinations of preprocessing methods and data partition methods were calibrated and validated. The model robustness during the calibration phase and validation phase would provide evidence for the effectiveness of the NIR spectrometer and the strategy to maximize spectrometer performance through preprocessing and data partition.

## 3. Results and Discussion

### 3.1. Descriptive Statistics of Spray Dried Whey Protein Matrix

The descriptive statistics of moisture content and protein content are shown in Table 2. The moisture content of the 21 set A samples ranges from 5.97 % to 8.76%, and the 20 set B samples range from 5.02% to 7.19%. The moisture content was in a typical range for spray-dried powers after three weeks of storage. For protein content, set A comprises samples formulated with a gradient of WPI from 10% to 25%, and the standard deviation of set A protein content reflects that trait. The protein content of sample set B, on the other hand, was designed to be distributed close to the mean value.

### 3.2. Comparison of Global Protein Model and Local Protein Model

For the comparison of global and local protein models, a total of 12 calibration models were established as shown in Table 3. To establish the local protein models that focus on a narrow distribution of protein content, 13 calibration samples were selected from set B, and the 6 remaining samples became the validation samples. Among the calibration samples in set A, 13 samples were also selected to establish global protein models, which were validated by the same validation samples used for their corresponding local models. All models except one (SNV, set B, KS, R^2^_C_ = 0.75) achieved good linear fitting in the calibration phase (R^2^_C_ > 0.90). In the validation phase, models established from set B have RMSEVs lower than 0.25%, and the RMSEVs range from 0.21% to 0.75%. Due to the relatively small variation in the protein content of set B (standard deviation = 0.56%) and small calibration sample size (13), the chemical information contained in the NIR spectra was more susceptible to instrumental noise and other interference, and thus the models resulted in poorer linear fitting to the validation samples and higher predication errors in the validation phase. Overall, across all models in the validation phase, the models based on set B samples demonstrated greater robustness with an average RMSEV of 0.17% compared to set A’s average RMSEV of 0.46%, which was expected since the local validation samples have similar characteristics to the local calibration samples. Similar results were also reported [36] where global models built on different temperatures underperform when compared to local models built on room temperature. However, in this study, one global model (first derivative, set A, SPXY) achieved an R^2^_V_ of 0.67 and RMSEV of 0.11%, which supported the hypothesis that a well-trained global protein model could predict the properties of spray-dried whey protein matrix with a narrow protein content distribution due to the linear relationship of the protein content and NIR spectrum.

Among the three preprocessing methods, two data partition methods, and two types of sample distribution (set A vs. set B), no single preprocessing method or data partition method resulted in improved overall robustness. This result supports the finding that the ability of preprocessing and data partition methods to reduce interference and select a representative calibration set depends on the sample characteristics, spectrometer, measuring environment, etc. [25]. However, some combinations of methods resulted in greater model robustness than others. Spectra of both set A and set B preprocessed with first derivative established models with the least RMSEV (0.11%) and the third least RMSEV (0.14%) when divided by SPXY, which might be a result of interference on the data partition methods. Based on the differences in samples in the instrumental response (X) space and/or dependent variable (y) space, where the distance between samples represents the difference in their chemical characteristics, data partition methods select samples that are vastly different from each other to generate a representative calibration set. However, the difference also includes interference such as instrumental noise, powder morphology, and particle size [21], which disrupts the function of data partition methods. After reducing the interference with the preprocessing, some data partition methods can distinguish samples with more accurate information and thus improve the robustness of predictive models. 

To further explore the robustness of a global protein sample set (A) in the characterization of a local sample set (B), all 21 set A samples were preprocessed with first derivative to establish a calibration model, which was then validated by all 20 set B samples including the outlier with 15.62% protein content. The prediction result is demonstrated in Figure 1. The model utilizes four latent variables and an R^2^_C_ of 1.00, RMSEC of 0.19%, R^2^_v_ of 0.73, and RMSEV of 0.27%. Although the linear fitting on the validation samples (R^2^_v_) is not as close as the calibration samples, the prediction error in the validation phase (RMSEV) was similar to the one in the calibration phase (RMSEC). Furthermore, linear fitting might not be a good measurement of robustness in this situation where the validation samples had a narrow protein content distribution and an outlier was present. The prediction results suggested that the global protein model can be used to characterize the samples with focused protein content. A similar strategy might also be viable in other focused protein content ranges depending on the application of the global model, and in calibration models for other chemical components whose relationship with NIR spectrum can be described by the Beer–Lambert Law.

#### Identification of Feature Bands

As shown in Figure 2, the variable importance in the projection (VIP) of the protein model established with all 21 set A samples provided insight into the feature wavelengths of protein content in the spray-dried samples. Thirteen peaks (1132, 1196, 1410, 1486, 1676, 1754, 1900, 1972, 2066, 2151, 2269, 2310, 2419 nm) with a VIP greater than 1 illustrated the feature bands of NIR spectra (first derivative) that played an important role in the prediction of protein content, which were corresponding to certain chemical vibrations associated with protein. The peak at 2151 nm has the highest VIP (3.18). As shown in Figure 3, the increase of WPI concentration in the powder clearly shifted the spectral trend at 2151 nm and 2066 nm. The peak of 2151 nm has been found to be a result of the combination of the N-H bend second overtone, C-H stretch, C=O stretch, N-H in-plane bend, and C-N stretch [37]. A study on corn protein [24] also pointed out the significance of this spectral band for protein content, as a few wavelengths from this region were sufficient to develop multiple linear regression models for the prediction of corn protein content. Other feature bands identified based on VIP are also associated with chemical vibrations in protein. The value of 2066 nm is associated with the C=O carbonyl stretch second overtone of primary amide, 1676 nm with the C-H stretch first overtone, 1900 and 1972 nm with the NH_2_ bend, and 1410 and 1486 nm with the N-H stretch first overtone [37]. The findings of protein feature bands provide a direction for the simplification of predictive models that can be employed for the processing control of spray-drying by a rapid characterization of spray-dried powders. It also solidifies the significance of those chemical vibrations for the characterization of protein in spray-dried whey protein.

### 3.3. Comparison of a Benchtop NIR Spectrometer and a Portable NIR Spectrometer for Moisture Calibration

As shown in Table 4, 18 moisture predictive models were established based on the spectra acquired from PEAK (1650 nm to 2400 nm) and CARY (950 nm to 2500 nm). The calibration phase applied three preprocessing methods (no preprocessing, first derivative, and SNV) and three data partitioning methods (systematic sampling (Sys), KS, and SPXY). Overall, CARY has shown a more robust performance across various combinations of preprocessing and data partition methods than PEAK. CARY models have good linear fittings for calibration with the lowest R^2^_C_ = 0.95, and this performance continued in the validation phase with the lowest R^2^_v_ = 0.87. In terms of the root mean square errors, CARY models achieved low errors in both calibration and validation phases, with RMSEC ≤0.23% and RMSEV ≤0.29%. The best combination of preprocessing and data partition, in terms of RMSEV, was first derivative coupled with SPXY, which resulted in the lowest RMSEV (0.12%) among all models. This combination was also the best for protein prediction based on CARY, as previously discussed, which supported the hypothesis that an optimal calibration strategy can be found for a certain NIR scanning condition [25]. Depending on the characteristics of the sample, the spectral scanning environment, and the design of the spectrometer, preprocessing coupled with data partitioning has the potential to dampen noises in relation to measurements and select a representative calibration set containing most chemical information in the NIR spectra. 

As for models based on PEAK spectra, the R^2^_C_ range from 0.87% to 0.97% and R^2^_v_ varies from 0.60% to 0.93%. A wider range compared to CARY models also appeared in RMSEC and RMSEV, ranging from 0.16 to 0.37 and 0.26 to 0.40, respectively. These differences can be explained by the use of a fiber optic cable and the glass layer in the light path that inflict more noise in PEAK spectra, while in CARY, the samples were loaded into the instrument with less environmental interference. The best combination of preprocessing and data partitioning method was surprisingly no preprocessing coupled with SPXY (R^2^_v_ = 0.93, RMSEV = 0.26%). 

The principle of preprocessing methods is essentially transforming the spectrum to remove the instrumental noise and baseline shifts. However, editing the information contained in the spectrum comes with a risk that valuable information could be lost in preprocessing [25,38] which has been found in other studies [8,21]. Furthermore, the use of the fiber optic cable and the glass wall in the light path added uncertainty to the performance of commonly used preprocessing methods. Although it is reported that silicone glass has a limited impact on NIR spectra below 2700 nm [39], the difference in the thickness of the glass layer in sample containers is still an interfering factor. In terms of data partitioning methods, SPXY once again showed advantages as all SPXY models had a lower RMSEV compared to their counterparts. Overall, the best-performing PEAK model reached a similar level of linear fitting to CARY models, which suggested that PEAK can be used as a rapid moisture characterization tool for spray-dried whey protein matrixes.

## 4. Conclusions

The results from this study indicated that it is possible to establish an NIR multivariate model based on a spray-dried whey protein sample sets with a wide distribution of protein content for the characterization of samples with a much smaller variance of protein content (R^2^_V_ = 0.73 and RMSEV = 0.27%). VIP identified the feature bands for protein based on the PLS model, and the band at 2151 nm was found to be significant for the prediction of protein content. The feasibility of a portable NIR spectrometer was validated by predicting the moisture content of spray-dried powders through a glass container and a fiber optic cable, where the best-performing PLS model was validated with an R^2^_V_ of 0.93 and RMSEV of 0.26%. This study found that the best combination of preprocessing and data partitioning methods for the benchtop NIR spectrometer was the first derivative coupled with SPXY, which resulted in a strong performance of the PLS calibration model for protein content and moisture content. Although the preprocessing methods evaluated in the study did not improve the performance of moisture models based on PEAK, SPXY models showed advantages in the validation phase. Finally, the portable NIR sensor with the wavelength range of 1650 nm to 2400 nm has potential for the real-time characterization of spray-dried powders.

## Figures and Tables

**Figure 1 foods-12-00467-f001:**
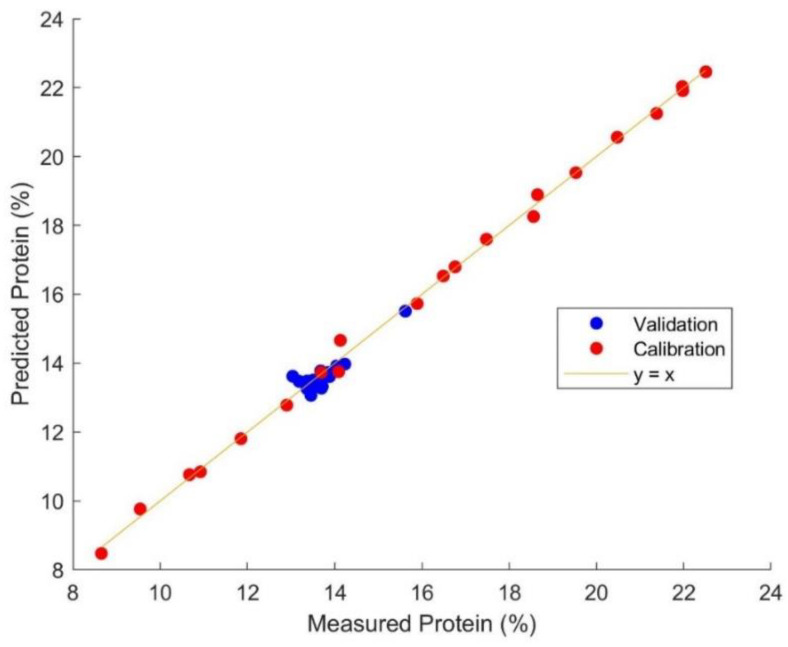
Prediction result of protein content of 20 local protein samples by a calibration model established with 21 global protein samples.

**Figure 2 foods-12-00467-f002:**
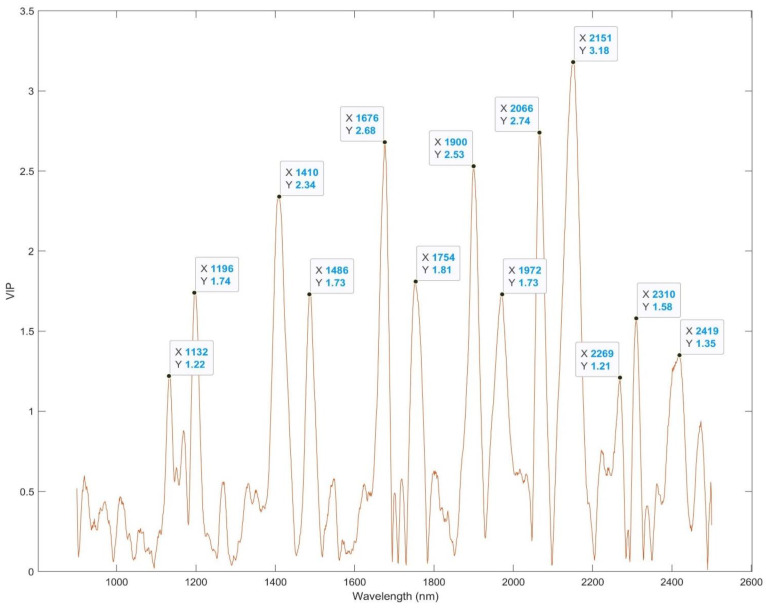
Variable importance in the projection (VIP) of the global protein model. X: wavelength; Y: VIP.

**Figure 3 foods-12-00467-f003:**
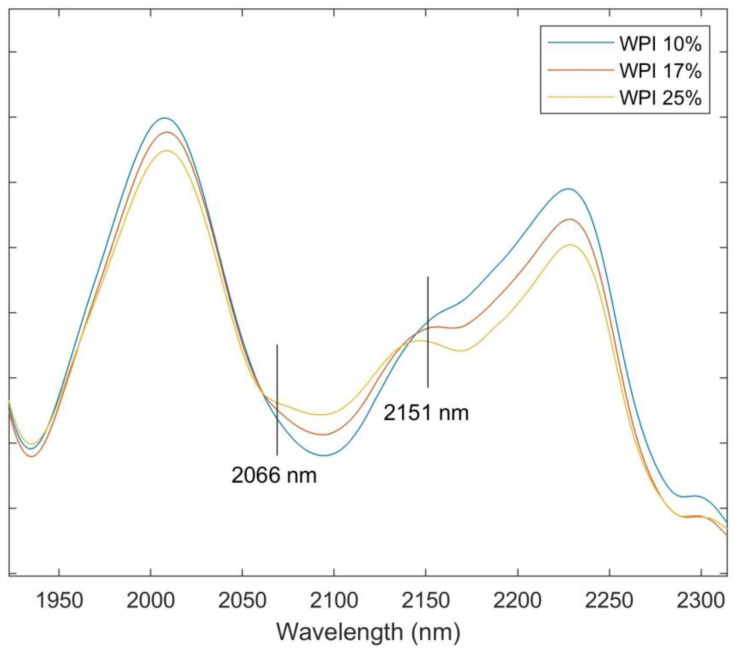
A section of spectra from samples formulated with three WPI concentrations (10%, 17%, and 25%), preprocessed with SNV and SG.

**Table 1 foods-12-00467-t001:** Composition of soli mixtures for the preparation of set A samples.

Sample Name	CAPSUL (g)	MD10 (g)	WPI (g)
C10	18.2	72.1	10.0
C11	17.8	71.2	11.0
C12	17.6	70.4	12.0
C13	17.4	69.6	13.1
C14	17.3	68.8	14.0
C15	17.0	68.3	15.3
C16	16.8	67.2	16.0
C17	16.6	66.4	17.1
C18	16.4	65.6	18.1
C19	16.2	64.8	19.0
C20	16.1	64.1	20.1
C21	15.8	63.2	21.1
C22	15.6	62.4	22.1
C23	15.4	61.6	23.1
C24	15.2	60.9	24.1
C25	15.1	60.0	25.0
V12	8.9	35.1	6.0
V15	8.5	34.1	7.6
V18	8.3	32.8	9.1
V21	7.9	31.6	10.5
V24	7.6	30.5	12.1

CAPSUL: modified starch; MD10: maltodextrin DE10; WPI: whey protein isolate.

**Table 2 foods-12-00467-t002:** Mean moisture content (w.b.) and protein content of two sample sets.

Sample Set	Mean Moisture Content (%, *w*/*w*)	Mean Protein Content (%, *w*/*w*)
A	7.35 ± 0.87	16.27 ± 4.32
B	6.01 ± 0.56	13.75 ± 0.52
A + B	6.66 ± 0.99	15.04 ± 3.36

**Table 3 foods-12-00467-t003:** Validation performance of global protein models (set A) and local protein models (set B).

Data Partition		KS					SPXY				
Preprocessing	Sample Set	#LV	R^2^_C_	RMSEC (%)	R^2^_V_	RMSEV (%)	#LV	R^2^_C_	RMSEC (%)	R^2^_V_	RMSEV (%)
None	A	6	0.99	0.39	−1.83	0.57	6	0.99	0.39	−1.86	0.41
	B	6	0.93	0.07	0.68	0.19	6	0.91	0.10	0.53	0.17
First derivative	A	4	1.00	0.09	−0.27	0.48	5	1.00	0.04	0.67	0.11
	B	7	1.00	0.01	0.37	0.25	7	1.00	0.01	0.48	0.14
SNV	A	6	1.00	0.06	−3.87	0.52	5	1.00	0.14	−13.98	0.69
	B	4	0.75	0.16	0.75	0.12	5	0.91	0.10	0.21	0.16

#LV: number of latent variables employed in the model; R^2^_C_: coefficient of determination for calibration; RMSEC: root mean square error of calibration; R_2_^V^: coefficient of determination for validation; RMSEV: root mean square error of validation.

**Table 4 foods-12-00467-t004:** Validation performance of moisture predictive models established based on a portable NIR spectrometer (PEAK) and a benchtop spectrometer (CARY).

Spectrometer	Preprocessing	Data Partition	#LV	R^2^_C_	RMSEC (%)	R^2^_V_	RMSEV (%)
PEAK	None	Sys	4	0.92	0.30	0.82	0.39
		KS	4	0.93	0.25	0.87	0.39
		SPXY	4	0.88	0.33	0.93	0.26
	First derivative	Sys	3	0.96	0.21	0.85	0.36
		KS	3	0.95	0.22	0.21	0.41
		SPXY	3	0.93	0.28	0.81	0.28
	SNV	Sys	3	0.90	0.33	0.84	0.37
		KS	3	0.90	0.32	0.83	0.40
		SPXY	3	0.87	0.36	0.92	0.28
CARY	None	Sys	4	0.97	0.19	0.96	0.19
		KS	3	0.95	0.23	0.94	0.23
		SPXY	4	0.98	0.15	0.90	0.29
	1st derivative	Sys	3	0.98	0.15	0.96	0.18
		KS	4	0.98	0.14	0.95	0.16
		SPXY	3	0.97	0.18	0.94	0.12
	SNV	Sys	3	0.95	0.22	0.93	0.25
		KS	4	0.96	0.20	0.91	0.16
		SPXY	4	0.96	0.21	0.87	0.13

## Data Availability

The datasets generated for this study are available on request to the corresponding author.

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
