# Peer review of "Characterizing Spray-Dried Powders through NIR Spectroscopy: Effect of Two Preparation Strategies for Calibration Samples and Comparison of Two Types of NIR Spectrometers"

_foods, 2023, doi:10.3390/foods12030467_

Round 1
Reviewer 1 Report
This study aims at understanding the application of NIR spectroscopy in spray dried samples by comparing two sample preparation strategies and two spectrometers. The topic of the manuscript is interesting; however, the manuscript is extremely technical and of difficult comprehension to the readers.
As the article is difficult to follow, I would suggest that authors revise the manuscript to make clear to the reader what the purpose of the article is, and what is the added value of the manuscript. I suggest authors provide a brief explanation of what are calibration samples and why it is needed. Also, to provide further information on global and local set.
I would also suggest the title is shortened.
In the introduction, a lot of focus was given to the spray drying process and very little info was provided for NIR spectroscopy. Therefore, I suggest the authors include more info on the principles of NIR as this is the focus of the manuscript.
Other comments:
Line 57: Could the authors provide further info on how this technology would reduce waste?
Line 60: Authors claim there is no investigation on the characterization of WPI-maltodextrin matrix but how about on other similar matrixes?
Line 85: I would suggest authors provide a table with the combination of solid mixtures used
Line 92: Was any repetition carried out?
Line 137: Punctuation
Lines 158-161: This fits more to the introduction then to methodology
Line 205: Which period? Hours, days, weeks, months, year? Be specific
Reviewer 2 Report
This manuscript, which is entitled" Characterizing spray dried whey protein-maltodextrin matrix through NIR spectroscopy: effect of two preparation strategies for calibration samples and comparison of two types of NIR spectrometers" has valuable results.
I just recommend the authors review the figure and its footnotes, particularly in fig. 2,
please add the footnotes regarding (what is X, and Y, refer to???
Reviewer 3 Report
Using two different NIR spectroscopy, the researchers worked on characterizing spray-dried whey protein maltodextrin matrix. It is a well-written paper and an interesting subject; comments below should be replied to, and the paper should be modified accordingly.
• Line 110. Section 2.6. Even though the NIR can penetrate from the glass jar, it is still important to provide the brand and specification of the jar. This is also the same for the Line 123 diffuse reflectance accessory. Please provide the manufacturer's info.
• Table 1. Why does A+B not have mean protein content?
• Table 2. Are the R2V values correct? The numbers look wrong to me.
• How did the researchers choose the calibration samples and validation samples, randomly or not? If it was randomly done, did they use any auto selection, or did they choose manually?
Round 2
Reviewer 1 Report
The article has improved however I am concerned about the lack of repetitions. I asked in the previous revision if any replicates were done and they replied no. Each of the experimental conditions should have been replicated at least 3 times to guarantee the statistics of the sample can well represent the parameters of the population. Without any repetition how can the authors guarantee the results are reliable?
Other minor comments:
Table 1. solid